# Influence of *ABO* Locus on PFA-100 Collagen-ADP Closure Time Is Not Totally Dependent on the Von Willebrand Factor. Results of a GWAS on GAIT-2 Project Phenotypes

**DOI:** 10.3390/ijms20133221

**Published:** 2019-06-30

**Authors:** Núria Pujol-Moix, Angel Martinez-Perez, Maria Sabater-Lleal, Dolors Llobet, Noèlia Vilalta, Anders Hamsten, Joan Carles Souto, José Manuel Soria

**Affiliations:** 1Thrombosis and Hemostasis Research Group, Institute of Biomedical Research (IIB-Sant Pau), 08025 Barcelona, Spain; 2Department of Medicine, Universitat Autònoma de Barcelona, 08025 Barcelona, Spain; 3Unit of Genomics of Complex Diseases, Institute of Biomedical Research (IIB-Sant Pau), 08025 Barcelona, Spain; 4Cardiovascular Medicine Unit, Department of Medicine, Center of Molecular Medicine, Karolinska Institutet, SE-17177 Stockholm, Sweden; 5Unit of Hemostasis and Thrombosis, Hospital de la Santa Creu i Sant Pau, 08025 Barcelona, Spain

**Keywords:** platelet reactivity, platelet function test, *ABO* blood-group system, von Willebrand factor, factor VIII

## Abstract

(1) Background: In a previous study, we found that two phenotypes related to platelet reactivity, measured with the PFA-100 system, were highly heritable. The aim of the present study was to identify genetic determinants that influence the variability of these phenotypes: closure time of collagen-ADP (Col-ADP) and of collagen-epinephrine (Col-Epi). (2) Methods: As part of the GAIT-2 (Genetic Analysis of Idiopathic Thrombophilia (2) Project, 935 individuals from 35 large Spanish families were studied. A genome-wide association study (GWAS) with ≈ 10 M single nucleotide polymorphisms (SNPs) was carried out with Col-ADP and Col-Epi phenotypes. (3) Results: The study yielded significant genetic signals that mapped to the *ABO* locus. After adjusting both phenotypes for the *ABO* genotype, these signals disappeared. After adjusting for von Willebrand factor (VWF) or for coagulation factor VIII (FVIII), the significant signals disappeared totally for Col-Epi phenotype but only partially for Col-ADP phenotype. (4) Conclusion: Our results suggest that the *ABO* locus exerts the main genetic influence on PFA-100 phenotypes. However, while the effect of the *ABO* locus on Col-Epi phenotype is mediated through VWF and/or FVIII, the effect of the *ABO* locus on Col-ADP phenotype is partly produced through VWF and/or FVIII, and partly through other mechanisms.

## 1. Introduction

Platelet reactivity can be measured using a wide variety of laboratory functional tests. Such tests are classified into two main types [1]: (1) tests based on platelet aggregation activated by agonists, in platelet-rich plasma or in whole blood; (2) tests based on platelet adhesion under shear stress. Among the second group of tests, the PFA-100 system (Siemens Healthcare Diagnostics, Marburg, Germany) measures platelet function by simulating in vitro a vessel wall under shear stress. The vessel wall is simulated by a membrane coated with collagen; it is coated also with ADP (cartridge collagen-ADP) or with epinephrine (cartridge collagen-epinephrine) as platelet agonists. The membrane has a hole through which the anticoagulated blood passes; the closure times (CTs) of this hole are inversely proportional to the functional capacity of platelets. The PFA-100 system was introduced in 1995 with the objective of measuring primary hemostasis in whole blood in vitro as a non-invasive and more accurate method than the bleeding time in vivo [2,3]. Long CTs gave reliable measurements of hemostatic deficiencies due to low levels of von Willebrand factor (VWF) or platelet functional defects [2,4]. The use of PFA-100 was subsequently expanded to the assessment of the effect of platelet hyperreactivity, as indicated by shortened CTs, on arterial and venous thrombotic risk [3,5,6,7,8]. Citrate concentration, platelet count, hematocrit, white blood cell count, circadian rhythm and some dietary elements can influence the CTs [3,9,10]. Age and sex do not have appreciable influence although slight shortening of CTs has been described in neonates, in children and in elderly men [3,10,11]. Obviously, VWF has an important influence on CTs, which correlates inversely with VWF levels [3,4,9,10]. Also, the *ABO* blood group influences PFA-100 CTs, with O group individuals having longer CTs than those of non-O groups [3,4,9,10,12,13]; this has been interpreted as an effect of the lower levels of VWF in group O individuals [4,9,10]. Some authors have suggested that factor VIII (FVIII) has no impact on the PFA-100 CTs [2,5], while others have found the contrary [14].

The GAIT-2 (Genetic Analysis of Idiopathic Thrombophilia (2) Project is a family-based genetic study designed to identify new genetic markers of thrombotic risk [15]. Using the variance component statistical method, the heritability of intermediate phenotypes that could play a role in thrombotic risk was determined. In the GAIT-2 Project, two PFA-100 phenotypes were included as a measure of platelet reactivity: CTs of collagen-ADP cartridge (Col-ADP) and of collagen-epinephrine cartridge (Col-Epi). Both phenotypes showed a strong genetic component with a heritability of 0.45 for Col-ADP and 0.52 for Col-Epi [16].

The objectives of the present study were: (1) to analyze the genetic correlations between Col-ADP and Col-Epi phenotypes with each other and with other related phenotypes, and (2) to perform a genome-wide association study (GWAS) to identify susceptibility loci for Col-ADP and Col-Epi phenotypes.

## 2. Results and Discussion

### 2.1. Genetic Correlations of the PFA-100 Phenotypes

The genetic correlations of Col-ADP and Col-Epi with each other and with the VWF antigen, coagulant FVIII and *ABO* genotype (considering dominant effect of allele *O*) are presented in Table 1.

As expected, the genetic correlations of PFA-100 phenotypes with VWF and *ABO* blood group were statistically significant. In addition, a significant correlation between the PFA-phenotypes and FVIII was also observed. Unlike VWF, which has an important role in primary hemostasis, FVIII is fundamental for coagulation. The mechanism by which the FVIII influences the PFA-100 CTs could be partially explained by its close relationship with the VWF. Both factors circulate together, and their levels are related. Moreover, genetic studies have demonstrated that there is a huge overlap between genetic factors regulating FVIII and VWF [17,18].

### 2.2. GWAS of the PFA-100 Phenotypes

Manhattan plots of the GWAS on Col-ADP and Col-Epi phenotypes are shown in Figure 1.

For each phenotype, a signal on chromosome 9 at the *ABO* locus reached a genome-wide significance level (*p*-value < 5 × 10^−8^). Seventy single nucleotide polymorphisms (SNPs) were found to be associated with Col-ADP, with *p*-values up to 1.50 × 10^−15^. Of these 70 SNPs, 45 were associated also with Col-Epi but with less statistical significance: *p*-values up to 1.35 × 10^−9^. The complete list of these SNPs is described in Appendix A. For both Col-ADP and Col-Epi phenotypes, all significant SNPs were located within the *ABO* gene and in its adjacent intergenic region.

To the best of our knowledge, our study is the first GWAS of PFA-100 phenotypes. Previous GWAS reports of platelet reactivity phenotypes were mainly based on aggregometry. Using GWAS and other genetic approaches, several SNPs have been found in association with platelet aggregometry phenotypes. These were located in different genes related to platelet functional receptors, regulators of cytoskeleton and signaling proteins. Among these genes, the following were described: *MIR100HG*, *MME*, *PIP3-E*, *GLIS3*, *LDHAL6A*, *ANKS1B*, *PIK3CG*, *MAGI1*, *C8orf86*, *FGFR1*, *LPAR1*, *CACNB2*, *SLC39A12*, *RPP25*, *SCAMP5*, *BMPR1A* (revisited in Bunimov et al. [19]), *ANKRD26* [20], *pannexin* [21], *ADRA2* [22] and the most relevant *PEAR-1* [22,23,24]. Notably, none of these genetic variants was found in the *ABO* locus. On the other hand, in our study, we did not find any of the SNPs previously reported to be associated with platelet aggregometry. To explain these different results, we should keep in mind that aggregometry, both in platelet-rich plasma and in whole blood, analyzes aggregation, that is, the platelet-platelet binding mediated by fibrinogen. In contrast, the PFA-100 test basically analyzes adhesion under shear stress, which depends mainly on binding with VWF [2,10].

Nineteen of the SNPs that we found in the *ABO* locus have been previously reported in relation to thrombotic-related conditions (Table 2): Venous thromboembolism [25,26,27,28,29,30], myocardial infarction [31], large-vessel and cardioembolic stroke [32], large-artery arteriosclerosis [33], coronary artery disease [34,35] and coronary artery disease shared with venous thromboembolism [36]. The association of these SNPs with thrombosis and with the PFA-100 phenotypes suggests that this functional platelet test may be useful to estimate thrombotic risk, although further studies are needed to confirm this. Moreover, some of the SNPs have been described also in association with variations in VWF, FVIII, and/or variations in biological factors that may play indirect roles in thrombosis, such as adhesion molecules [17,18,37,38,39].

After finding a genetic correlation between the PFA-100 phenotypes and VWF, FVIII and *ABO* genotype, we adjusted the GWAS results for these factors. As shown in Figure 2 and Appendix A, after adjusting the Col-ADP results for VWF 50, SNPs remained significant but with lower significance than before adjustment (*p*-values up to 1.55 × 10^−10^). After adjusting the Col-ADP, the results for FVIII 46 SNPs remained significant; almost all were the same as for the VWF adjustment, but 6 were different. The remaining SNPs also had a lower significance than before (*p*-values up to 5.00 × 10^−11^).

It should be noted that after adjustment of Col-ADP results for *ABO* genotype, the association signal disappeared completely (data not shown). Regarding Col-Epi, the significant SNPs disappeared with any of the adjustments performed, including VWF, FVIII and *ABO* genotype.

### 2.3. Genetic Influence of ABO Locus on PFA-100 Phenotypes

Our GWAS results and adjustments suggested that the *ABO* gene was the main determinant of variations in PFA-100 CTs since all the significant SNPs associated with PFA-100 phenotypes were at the *ABO* locus and all of them lost their genome-wide significance when we adjusted for the *ABO* genotype.

However, there were differences between Col-ADP and Col-Epi phenotypes. With Col-Epi, all significant SNPs also disappeared when adjusted for VWF or FVIII. This suggested that the influence of the *ABO* gene was related mainly to these factors. Regarding Col-ADP, the adjustment either by VWF or by FVIII reduced the number of significant SNPs in both cases but 50 and 46 respectively did not disappear. This suggested that part of the genetic effect of the *ABO* gene on the Col-ADP was mediated by VWF and/or FVIII. The relationships among *ABO* blood group, VWF and FVIII have been previously described. Non-O group individuals have 25% higher levels of VWF and FVIII than O group individuals [37,38]. These differences are attributed to the formation of A and B antigens, catalyzed by specific glucosyltransferases, on the H antigen existing in VWF. They are related to the glycosylation and clearance rate of VWF which is lower in non-O individuals. We previously showed that FVIII and VWF were genetically correlated with thrombotic risk, and demonstrated significant linkage between the *ABO* locus and plasma levels of VWF and FVIII [40]. More recent GWAS demonstrated that *ABO* gene is by far the major determinant of VWF and FVIII levels [18]. This can explain the genetic relationship between the PFA-100 phenotypes and the *ABO* blood group through the VWF and/or FVIII.

According to our results, part of the effect of the *ABO* locus on the Col-ADP phenotype does not occur due to WVF and/or FVIII. Some studies have suggested alternative mechanisms by which the *ABO* gene can intervene in platelet reactivity, thrombosis, and cardiovascular diseases. Non-O individuals have higher levels of cholesterol [41]. Also, they have increased amounts of the adhesion molecules and cytokines involved in inflammation that are related to cardiovascular disease [38,39,42]. Ten of the SNPs that we found at the *ABO* locus, associated with PFA-100 phenotypes, have been described previously in association with adhesion molecules and/or cholesterol (Table 2).

Moreover, the *ABO* blood group can influence platelet reactivity through various mechanisms related to glycosylation [37,43]. A and B antigens expressed on platelet glycoproteins can modify the activity of the glycoprotein complexes involved in platelet adhesion and the galectin-glycan interactions; galectins from extracellular matrices are potent platelet agonists. A and B antigens, also present on platelet glycosphingolipids, are involved in platelet aggregation and thrombosis by binding cell adhesion molecules.

Glycosylation is essential also for other platelet functions in which no relationship with the *ABO* blood group has been described so far. An example of this is the glycosylation of the P2Y12 receptor [44]. P2Y12 is a G_i_-coupled ADP receptor that contains two potential N-linked glycosylation sites at its extracellular amino-terminus. The lack of glycosylation of this receptor leads to a defective P2Y12-mediated inhibition of the adenylyl cyclase activity resulting in defective platelet reactivity. The fact that the effect of the *ABO* locus, independent of the VWF and/or FVIII, was demonstrated only for the Col-ADP phenotype gives rise to speculation about the possibility that the *ABO* locus, among other mechanisms, could influence the glycosylation of the P2Y12 receptor. Further data are required to support this hypothesis.

## 3. Methods

### 3.1. Enrollment of Individuals and Families

The enrollment of individuals and families was described in detail in a previous publication [16]. Briefly, 935 individuals from 35 large Spanish families, included in the GAIT-2 Project, were recruited through a proband with idiopathic thrombophilia and the condition of having at least of 10 members in at least 3 generations willing to participate in the study. The exclusion criteria were: Deficiencies of antithrombin, protein S, protein C, heparin cofactor II, or plasminogen, activated protein C resistance, Factor V Leiden, dysfibrinogenemia, lupus anticoagulant and antiphospholipid antibodies. The subjects were questioned about their current medication to confirm that they had not taken antiplatelet drugs in the last two weeks, or other drugs with slight effect on platelet function (such as nonsteroidal anti-inflammatory drugs or serotonin reuptake inhibitor drugs) in the last week.

Among the individuals studied, 465 were male and 470 were female. The mean age was 39.5 (minimum 2.6, maximum 101, SD 21.4), and 197 of them were 18-years of age or younger. There were 86 with venous thrombosis, 47 with arterial thrombosis, and 13 with both venous and arterial thrombosis.

The study was performed according to the Declaration of Helsinki. Written informed consent was obtained from all adult patients and from parents or guardians of children. All procedures were approved by the Institutional Review Board at the Hospital de la Santa Creu i Sant Pau. The GAIT-2 Project was approved on November 23, 2005. At that time, no number was assigned to the approved projects.

### 3.2. Blood Collection, Laboratory Analyses and DNA Preparation

Whole blood samples were obtained by venipuncture, under basal conditions, after a 12 h overnight fast, and between 9:00 a.m. and 9:30 a.m. to minimize the circadian fluctuation. A 5 mL sample was obtained in EDTA-K3 for determining standard blood cell counts. A 5 mL sample of blood was collected in 3.8% sodium citrate to be analyzed in the PFA-100 device; the phenotypes obtained were: CTs (in seconds) for the cartridge collagen-ADP (Col-ADP phenotype) and for the cartridge collagen-epinephrine (Col-Epi phenotype). To avoid erroneous results due to thrombocytopenia or anemia, 6 individuals with platelet counts down to 100 × 10^9^/L and 8 individuals with hemoglobin down to 110 g/L were excluded for the of PFA-100 measurements. Another sample collected in 3.8% sodium citrate was used to obtain platelet-poor plasma by centrifugation at 2000 *g* for 20 min at room temperature. This plasma was used for determining VWF antigen by ELISA and coagulant FVIII activity as previously described [40]. VWF antigen and coagulant FVIII were recorded as percentages of an international standard sample. PFA-100 and FVIII activity assays were performed on fresh samples.

DNA was extracted from EDTA blood samples using a standard salting-out procedure [45] or a commercial kit (Wizard, Promega Corp, Madison, WI, USA). The *ABO* genotype, that distinguishes the *A*_1_, *A*_2_, *B*, *O*_1_ and *O*_2_ alleles, was determined as previously described [46]. The primers used were described in Souto et al. [40].

### 3.3. Genotyping Filtering and Imputation

We genotyped the samples from 934 individuals, with a combination of HumanOmniExpressExome-8v1.2 (324 individuals and coverage 964,193 variants) and HumanCoreExome-12v1.1 (610 individuals and coverage 542,585 variants). After filtering the datasets by call rate (>98%), Hardy-Weinberg Equilibrium (*p*-value > 10^−6^) and minor allele frequency (MAF) (>1%) and deleting the Mendelian errors, we obtained 395,556 SNPs in all of the samples. We estimated haplotypes using SHAPEIT v2 [47] and imputed genotypes to the 1000 genomes phase 1 panel using IMPUTE2 [48]. We imputed 37,985,264 SNPs, of which 10,844,567 remained after filtering again by MAF (>0.5%). The 14 individuals who had thrombocytopenia or anemia were excluded from the association analysis.

### 3.4. Statistical Correlation Analyses

The correlations among the phenotypes Col-ADP and Col-Epi with each other and, by pairs, with other biologically-related phenotypes, were analyzed by multivariate variance component models, which are an extension of the univariate model [49]. By studying these traits in extended families, we were able to estimate robustly both the genetic (ρ_g_), and the environmental (ρ_e_) correlations between pairs of traits.

### 3.5. GWAS of PFA-100 Phenotypes

We used two phases for the association analysis of normal transformed quantitative phenotypes with the imputed genotypes. The first was a fast screen of variants with Matrix eQTL [50] using modelLINEAR. The GWAS of each phenotype was calculated adjusting by age. Sex, smoking and and oral contraceptives were not included as covariates, since they did not influence the PFA-100 CTs in the GAIT-2 sample [16]. The “errorCovariance” parameter was used to account for the pedigree effect. Variants with a *p*-value < 10^−3^ were recalculated in the second phase with SOLARIUS [51] and SOLAR-Eclipse v 8.1.1 [52], using variance component methods. Only the *p*-values calculated with SOLAR are reported, since it is considered a gold standard in the field of extended pedigree samples. SOLAR employs the maximum likelihood approach for variance component models with the standard likelihood ratio tests to evaluate the statistical significance of the model’s parameters [53].

## 4. Conclusions

We report the first GWAS on PFA-100 phenotypes. The results of our study suggest that the *ABO* locus is the main determinant of these phenotypes and, in the case of Col-Epi, all the *ABO* influence is mediated by VWF and/or FVIII. However, for Col-ADP, the influence of *ABO* is only partially mediated by VWF and/or FVIII suggesting that other mechanisms may explain the *ABO* effect on this phenotype. Much remains to be done to better understand the relationships between *ABO* blood group and platelet reactivity measured by PFA-100 test, beyond those mediated by VWF and/or FVIII. Unraveling these mechanisms will help to identify novel pathways involved in platelet reactivity.

## Figures and Tables

**Figure 1 ijms-20-03221-f001:**
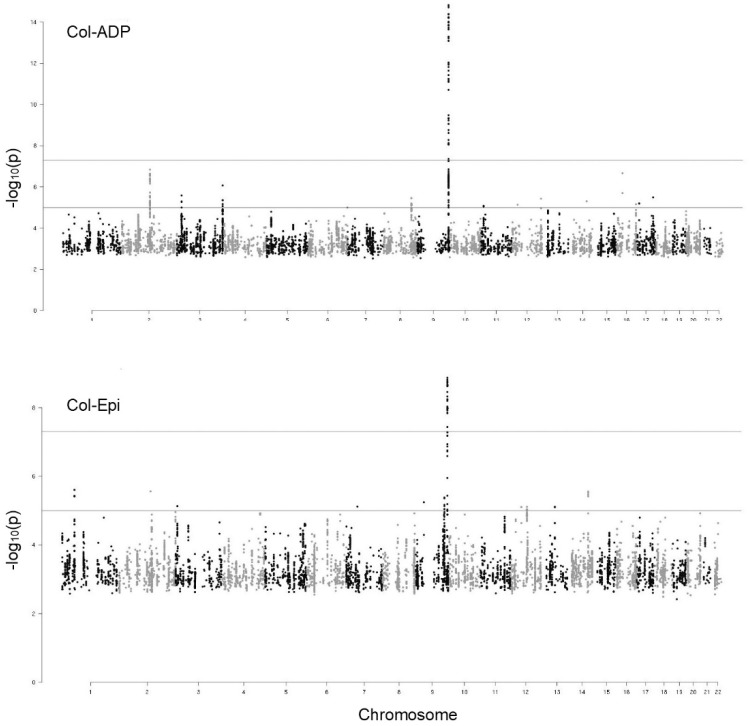
Manhattan plots of the GWAS on two PFA-100 phenotypes: collagen-ADP and collagen epinephrine closure times. Dots correspond to SNPs organized by chromosomal order and position and the vertical axis shows the statistical significance expressed as −log_10_ of the *p*-values. The horizontal lines mark the 5 × 10^−8^
*p*-value threshold of genome-wide significance.

**Figure 2 ijms-20-03221-f002:**
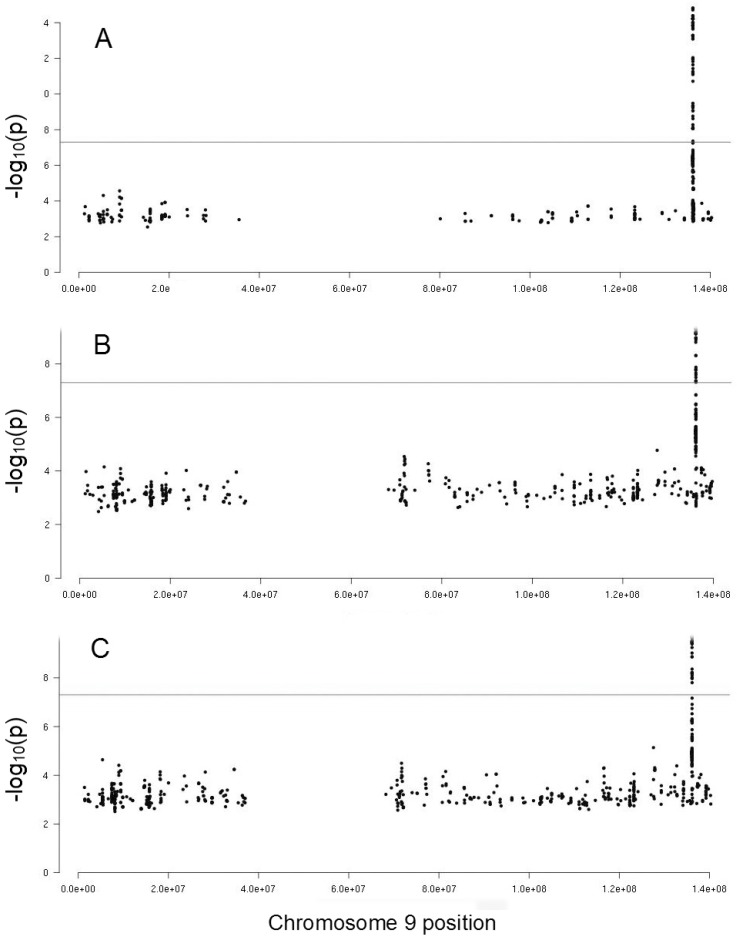
Manhattan plots of the chromosome 9 region of the GWAS on collagen-ADP closure time phenotype: (**A**) without adjustments, (**B**) after adjusting for von Willebrand factor, and (**C**) after adjusting for factor VIII. Dots correspond to SNPs organized by position and the vertical axis shows the statistical significance expressed as -log_10_ of the *p*-values. The horizontal lines mark the 5 × 10^−8^
*p*-value threshold of genome-wide significance.

**Table 1 ijms-20-03221-t001:** Genetic correlations among PFA phenotypes, von Willebrand factor, coagulation factor VIII and *ABO* genotype.

Phenotype	Col-Epi	VWF	FVIII	*ABO*
**Col-ADP**	ρ = 0.7917 (5.80 × 10^−9^)	ρ = −0.7002 (1.02 × 10^−10^)	ρ = −0.6209 (2.66 × 10^−8^)	ρ = 0.5895 (7.01 × 10^−9^)
**Col-Epi**	-	ρ = −0.6342 (7.14 × 10^−8^)	ρ = −0.5947 (3.97 × 10^−7^)	ρ = 0.4477 (0.0003)

ρ = genetic correlation and *p*-value (in brackets); VWF = von Willebrand factor antigen; FVIII = coagulation factor VIII activity; *ABO* = *ABO* genotype, considering dominant effect of allele *O.*

**Table 2 ijms-20-03221-t002:** *ABO* locus (chromosome 9): SNPs associated with PFA-100 phenotypes which have been previously described in association with thrombosis-related conditions and with variations of biological factors.

SNP	Position (bp)	Location	MAF	Association with Col-ADP *p*-Value	Association with Col-Epi *p*-Value	Association with Thrombosis-Related Conditions [References]	Association with Variations of Biological Factors [References]
rs8176719	136132908	coding, 5-UTR, intron	0.459	5.21 × 10^−14^	1.88 × 10^−9^	VTE [18,25,26,29]	
rs687621	136137065	intron	0.432	1.49 × 10^−15^	3.50 × 10^−9^	VTE [18,26], MI [31], LVCES [32]	VWF [17,18,38], ICAM-1 [38]
rs687289	136137106	intron	0.433	1.93 × 10^−15^	5.72 × 10^−9^	MI [31], LVCES [32]	FVIII [16,17], ICAM-1 [38]
rs2519093	136141870	intron	0.312	1.16 × 10^−12^	6.01 × 10^−9^	VTE [25], LVCES [32]	-
rs514659	136142203	intron	0.433	6.04 × 10^−15^	1.04 × 10^−8^	VTE [26], MI [31], LVCES [32], LAA [33]	VWF [38]
rs644234	136142217	intron	0.460	1.42 × 10^−14^	2.18 × 10^−9^	MI [31], LVCES [32]	E-selectin [38]
rs643434	136142355	intron	0.460	1.43 × 10^−14^	2.18 × 10^−9^	MI [31], LVCES [32]	-
rs545971	136143372	Intron	0.433	6.03 × 10^−15^	1.04 × 10^−8^	MI [31], LVCES [32]	-
rs612169	136143442	intron	0.433	6.05 × 10^−15^	1.04 × 10^−8^	MI [31], LVCES [32]	ICAM-1, E-selectin [38]
rs674302	136146664	intron	0.433	6.06 × 10^−15^	1.04 × 10^−8^	MI [31], LVCES [32]	-
rs500498	136148647	intron	0.408	5.56 × 10^−08^	-	VTE [27], LVCES [32]	ICAM-1, E-selectin [38]
rs505922	136149229	intron	0.433	9.68 × 10^−15^	1.12 × 10^−8^	VTE [27,28], MI [31], LVCES [32]	-
rs529565	136149500	intron	0.434	1.86 × 10^−14^	1.01 × 10^−8^	VTE [29,30], MI [31], LVCES [32], LAA [33]	-
rs630014	136149722	intron	0.407	8.74 × 10^−10^	-	VTE [27,28], LVCES [32]	E-selectin [38]
rs651007	136153875	intergenic	0.204	8.02 × 10^−12^	2.32 × 10^−9^	LVCES [32], LAA [33], CAD [34]	VWF, ICAM-1, E-selectin, cholesterol [38]
rs579459	136154168	intergenic	0.340	3.75 × 10^−12^	1.35 × 10^−9^	LVCES [32], CAD [35], CAD+VTE [36]	ICAM-1, E- and P-selectin [38]
rs649129	136154304	intergenic	0.338	6.88 × 10^−12^	2.19 × 10^−9^	LVCES [32]	ICAM-1, LDL-cholesterol [38,39]
rs495828	136154867	intergenic	0.340	7.71 × 10^−12^	2.28 × 10^−9^	VTE [25,27], LVCES [32]	ACE [38]
rs633862	136155444	intergenic	0.391	2.41 × 10^−9^	-	LVCES [32]	-

VTE = venous thromboembolism; MI = myocardial infarction; LVCES = large-vessel and cardioembolic stroke; LAA = large-artery arteriosclerosis; CAD = coronary artery disease; CAD + VTE = coronary artery disease shared with venous thromboembolism; VWF = von Willebrand factor; FVIII = coagulant factor VIII; ACE = angiotensin-converting-enzyme.

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
