# Peer review of "Influence of ABO Locus on PFA-100 Collagen-ADP Closure Time Is Not Totally Dependent on the Von Willebrand Factor. Results of a GWAS on GAIT-2 Project Phenotypes"

_ijms, 2019, doi:10.3390/ijms20133221_

Round 1

Reviewer 1 Report

The authors present some novel findings. There are occasional gaps that can be filled; specifically:

Page 2: “Although coagulation factor FVIII (FVIII) circulates together with VWF and the levels of both factors are related, variations of FVIII affecting PFA-100 CTs have not been described [2,5].” Not entirely true: refer to [Favaloro EJ, Thom J, Patterson D, Just S, Baccala M, Dixon T, Meiring M, Koutts J, Rowell J, Baker R. Potential supplementary utility of combined PFA-100 and functional VWF testing for the laboratory assessment of desmopressin and factor concentrate therapy in von Willebrand disease. Blood Coag Fibrinolysis, 2009; 20:475–483.] where a clear relationship is shown between FVIII and PFA CTs

P7: Was only VWF:Ag performed? No VWF activity assays? The relationship of VWF with PFA CTs is strongest with functional assays such as collagen binding. May need statement of limitations.

Some prior publications assessing PFA CTs with thrombosis risk seem to be missing from the paper; suggest authors check [Favaloro EJ. Clinical Utility of the PFA-100. Semin Thromb Hemost 2008; 34:709-733] for a review of publications linking short CTs with thrombosis risk, where it was also identified that C/ADP (not C/Epi) seemed most relevant.

Author Response

Response to Reviewer 1 Comments

Point 1: The authors present some novel findings. There are occasional gaps that can be filled; specifically:

Page 2: “Although coagulation factor FVIII (FVIII) circulates together with VWF and the levels of both factors are related, variations of FVIII affecting PFA-100 CTs have not been described [2,5].” Not entirely true: refer to [Favaloro EJ, Thom J, Patterson D, Just S, Baccala M, Dixon T, Meiring M, Koutts J, Rowell J, Baker R. Potential supplementary utility of combined PFA-100 and functional VWF testing for the laboratory assessment of desmopressin and factor concentrate therapy in von Willebrand disease. Blood Coag Fibrinolysis, 2009; 20:475–483.] where a clear relationship is shown between FVIII and PFA CTs

Response 1: The article recommended by the reviewer [Favaloro et al. Blood Coag Fibrinolysis, 2009; 20:475–483] shows a clear relationship between factor VIII and PFA-100 CTs. However, some articles on PFA-100 do not describe any effect of factor VIII on PFA CTs.

In the revised versions of the manuscript, two paragraphs on pages 2 and 3 have been changed (lines 57 to 62, and 86 to 96) according to the comments of the reviewer. The recommended reference has been added.

Point 2: P7: Was only VWF:Ag performed? No VWF activity assays? The relationship of VWF with PFA CTs is strongest with functional assays such as collagen binding. May need statement of limitations.

Response 2: We agree that the VWF activity assays, such as collagen binding, better demonstrate the influence of the VWF on PFA-100 CTs. However, the number of determinations made to the individuals in the GAIT-2 Project had limitations.

The GAIT-2 Project included samples for different types of phenotypes, including, among others, whole blood counts, platelet parameters, coagulation factors, platelet and leukocyte flow cytometry, cytokines and adhesion molecules, and platelet electron microscopy. Also, we collected samples for DNA extraction and plasma samples to be frozen for later use to determine more phenotypes. This implied a limitation in the number of analyzes that needed to be performed immediately or within the first hours after the blood collection.

Point 3: Some prior publications assessing PFA CTs with thrombosis risk seem to be missing from the paper; suggest authors check [Favaloro EJ. Clinical Utility of the PFA-100. Semin Thromb Hemost 2008; 34:709-733] for a review of publications linking short CTs with thrombosis risk, where it was also identified that C/ADP (not C/Epi) seemed most relevant.

Response 3: The authors already knew the article suggested by the reviewer [Favaloro et al, Semin Thromb Hemost 2008; 34:709-733] but we considered it sufficient to cite a more recent article by the same author [Favaloro et al, Am J Hematol 2017; 92: 398-404] including the assessment of PFA CTs with thrombosis risk and citing the suggested article. However, after reading again this article of 2008, we find out that actually all aspects of PFA-100 related to thrombotic risk are better explained than in the 2017 paper. Therefore, in the revised version of the manuscript, we have added the suggested reference.

Reviewer 2 Report

Pujoi-Moix et al previously described two phenotypes; closure times to collagen-ADP and collagen-epinephrine in a PFA-100 system.  The ABO locus was found to be the main genetic determinant in the closure times.  This was completely mediated by vWF and FVIII in the collagen-epinephrine phenotype but on partially in the collagne-ADP phenotype.  Understanding the genetic determinants of platelet function could provide useful for detection of thrombotic risk.  

The study appears well-designed study and the manuscript is clear.  Some minor points for consideration are listed below;

I do not think the title best describes the main findings.  All of the effect of the ABO locus on the col-epi phenotype was attributed to vWF and FVIII yet the title suggests that only part of the influence of the ABO locus is dependent on vWF.  Please consider revising.

The authors did not find previously reported SNPs detected in aggregometry studies but did the authors compare the correlation of the SNPS detected in this study with other measures of platelet activation?  E.g. Platelet aggregometry, P-selectin and phosphatidylserine exposure? 

Did the authors perform a full blood count?  Did they account for differences in platelet number and haematocrit, which will influence the closure times.

The authors state that p-value threshold of genome-wide significance is 5 x 10-8 and that the correlation between ABO group and FVIII were significant but the p value with Col-Epi is greater than this threshold.  Please clarify the significance level.

Author Response

Response to Reviewer 2 Comments

Pujol-Moix et al previously described two phenotypes; closure times to collagen-ADP and collagen-epinephrine in a PFA-100 system.  The ABO locus was found to be the main genetic determinant in the closure times.  This was completely mediated by vWF and FVIII in the collagen-epinephrine phenotype but on partially in the collagne-ADP phenotype.  Understanding the genetic determinants of platelet function could provide useful for detection of thrombotic risk. 

The study appears well-designed study and the manuscript is clear.  Some minor points for consideration are listed below;

Point 1: I do not think the title best describes the main findings.  All of the effect of the ABO locus on the col-epi phenotype was attributed to vWF and FVIII yet the title suggests that only part of the influence of the ABO locus is dependent on vWF.  Please consider revising.

Response 1: Following the suggestion of the reviewer we have changed the title of the manuscript

Point 2: The authors did not find previously reported SNPs detected in aggregometry studies but did the authors compare the correlation of the SNPS detected in this study with other measures of platelet activation?  E.g. Platelet aggregometry, P-selectin and phosphatidylserine exposure?

Response 2: We agree that it would have been very interesting to perform other tests of platelet function such as aggregometry or platelet activation. However, the number of determinations made to the individuals in the GAIT-2 Project had limitations.

The GAIT-2 Project included samples for different types of phenotypes, including, among others, whole blood counts, platelet parameters, coagulation factors, platelet and leukocyte flow cytometry, cytokines and adhesion molecules, and platelet electron microscopy. Also, we collected samples for DNA extraction and plasma samples to be frozen for later use to determine more phenotypes. This implied a limitation in the number of analyzes that needed to be performed immediately or within the first hours after the blood collection.

Point 3: Did the authors perform a full blood count?  Did they account for differences in platelet number and haematocrit, which will influence the closure times.

Response 3: Yes, a full blood count was determined to all individuals in the study. Taking into account the influence on closure times, some individuals with slightly lower levels of hemoglobin or platelets were excluded from the study. In the revised version of the manuscript two paragraphs have been added explaining these exclusions: page 8  (lines 227 to 229, and 247 to 248).

Point 4: The authors state that p-value threshold of genome-wide significance is 5 x 10-8 and that the correlation between ABO group and FVIII were significant but the p value with Col-Epi is greater than this threshold.  Please clarify the significance level.

Response 4: We assume the reviewer refers to Supplementary Table S1. Indeed, some p-values (of Col-Epi, and also some of Col-ADP), are greater than the defined threshold of genome-wide significance. We are sorry, there are mistakes.

In the revised version of the manuscript the Table S1 has been corrected.